# Factors Influencing Preventive Behavior of COVID-19 among Nursing Students in South Korea

**DOI:** 10.3390/ijerph191912094

**Published:** 2022-09-24

**Authors:** Seung-Kyoung Yang, Minji Kim

**Affiliations:** Department of Nursing, Kyungnam University, 7, Gyeongnamdaehak-ro, Masanhappo-gu, Changwon-si 51767, Korea

**Keywords:** COVID-19, stress, perception, self-efficacy, students, nursing

## Abstract

We aimed to identify factors influencing the preventive behavior of COVID-19 among nursing students. A survey was conducted through an online questionnaire in December 2021 for students in the department of nursing at a university located in C city. An online questionnaire was distributed to 189 nursing students who voluntarily agreed to participate in the study, and data from 179 of them were used for the final analysis. The research tools used were COVID-19 stress scale for Korean people, COVID-19 risk-perception scale, self-efficacy scale, and COVID-19 prevention behavior scale. Data were analyzed by descriptive statistics, t-test, analysis of variance, Mann–Whitney and Kruskal–Wallis test, as well as Spearman correlation coefficient and multiple linear regression with SPSS 23.0 program. Factors significantly influencing the preventive behavior of COVID-19 were self-efficacy (β = 0.53, *p* < 0.001) and COVID-19 stress (β = 0.25, *p* = 0.001). The explanatory power of related variables was 45.0%. We found that self-efficacy and COVID-19 stress had a significant effect on the preventive behavior of nursing students. Therefore, to promote the preventive behavior of COVID-19, it is necessary to improve the self-efficacy of nursing students and manage COVID-19 stress well.

## 1. Introduction

The coronavirus disease (COVID-19), which started in December 2019 and was declared a pandemic by the World Health Organization on 11 March 2020, has spread rapidly worldwide and continues to be a global scourge [1]. As of July 15, 2022, there were a total of 538,737,767 confirmed cases of COVID-19, and the cumulative number of deaths exceeded 6,332,165 [2]. In South Korea, the cumulative numbers of confirmed cases and deaths were 18,761,757 and 24,742, respectively [2]. Measures to prevent infection include handwashing, wearing the correct mask, ventilation, disinfection, social distancing, and vaccination. In Korea, efforts are being made to prevent the spread of COVID-19 by washing hands, wearing the correct mask, ventilation, disinfection, social distancing, and vaccination [3]. South Korea has been implementing social distancing in stages to prevent infection and spread. Since April 2022, social distancing measures have been lifted for the first time in 2 years and 1 month due to the decreasing trend in the size of the epidemic, and the COVID-19 grade has been adjusted from Grade 1 to Grade 2 [4]. However, even after the social distancing curbs have been lifted, infection prevention by observing personal quarantine rules remains an important issue. In particular, as nursing students play an important role in preventing the spread of infection in communities and hospitals as premedical personnel [5], awareness of their COVID-19 preventive behaviors is important.

During the COVID-19 outbreak, the Ministry of Education recommended switching from face-to-face classes to non-face-to-face parallel and non-face-to-face classes according to the distancing steps [6]. Nursing students had higher levels of confusion, maladaptation, and stress than students pursuing other majors due to the suspension or temporary suspension of clinical practice at the hospital during the course of clinical practice, in addition to school teaching methods, and reinforcement of infection control in the training sites [7,8]. As such, the spread of COVID-19 had a direct adverse impact on nursing students compared to college students in other majors, and it is thought that students are likely to be exposed to various stresses in performing COVID-19 preventive behaviors. COVID-19 stress in college students, except for those in health departments, was a significant influencing factor for preventive health behavior [9]. However, no study has examined the relationship between COVID-19 stress and infection preventive behaviors of nursing students, who are known to have high levels of COVID-19-related stress; therefore, it would be meaningful to understand this relationship.

New infectious diseases are associated with high uncertainty about the treatment course, the disease burden on the public [10], and uncertainty and risk perception, which are interrelated [11]. Risk perception is a subjective evaluation of risk occurrence [12]. Risk perception at the individual level directly affects the preventive behavior compared to the risk perception at the social level [13]. The higher an individual’s perceived risk of COVID-19, the greater is the infection preventive behavior [14]. Therefore, it will be possible to increase infection preventive practices by understanding the degree of risk perception of COVID-19 among individuals.

To prevent the spread of a novel infectious disease, it is very important to institute active infection preventive behavior [15]. In particular, nursing students are easily exposed to infectious diseases in the medical field, and to prevent the spread of infectious diseases, proper infection preventive behavior should be emphasized [16]. Self-efficacy refers to the belief in one’s own ability to perform the actions that are necessary to achieve the goal [17], and was found to be a significant influencing factor in the preventive behavior intention and preventive behavior with regard to infectious diseases [18,19]. Self-efficacy was related to stress and health-related risk perception [20,21].

Although a study investigated the preventive behavior with regard to COVID-19 among nursing students [15,16,22], we found it challenging to identify any study that collectively investigated the variables of COVID-19 stress, risk perception, and self-efficacy. In this study, the association of COVID-19 stress, risk perception, and self-efficacy with COVID-19-related infection preventive behaviors of nursing students was identified to provide basic data for developing programs to promote infection preventive behaviors among nursing students.

## 2. Materials and Methods

### 2.1. Study Design

This study was performed to understand the effects of COVID-19 stress, risk perception, and self-efficacy in nursing students on the prevention of SARS-CoV-2 infection.

### 2.2. Participants

The participants were conveniently sampled from among students enrolled in the Department of Nursing at a university located in C city of Gyeongsangnam-do, Republic of Korea. Eligible participants were those who understood the purpose of the study and voluntarily agreed to participate in the study. The questionnaire survey was conducted using an online questionnaire. In this study, the nursing professor did not participate in the online survey, and the researcher distributed the online questionnaire to the students who voluntarily wanted to participate in the survey. The number of samples was calculated using the G*Power 3.1.9.2 program with a significance level of 0.05, power of 0.95, a medium effect size of 0.15, and 10 predictors in the multiple linear regression analysis. The minimum sample size required was 172 people, and the questionnaire was distributed to 189 people, considering a 10% dropout rate. After excluding respondents with insufficient responses, data from a total of 179 participants were used for the final analysis.

### 2.3. Instruments

#### 2.3.1. COVID-19 Stress

COVID-19 stress was measured using the COVID-19 Stress Scale for Korean People (CSSK), which was developed and validated by Kim et al. [23] to measure the kind of stress that Koreans experience due to COVID-19. The CSSK has a total of 21 items, consisting of 9 items of fear of infection, 6 items of difficulty due to social distancing, and 6 items of anger toward others. The scores are rated on a 5-point Likert scale, and a higher score means more stress in the relevant area. The coefficient alpha in the study of Kim et al. [23] and in our study were 0.91 and 0.89, respectively.

#### 2.3.2. Risk Perception

The COVID-19 Risk-Perception Scale developed by Taghrir et al. [24] for medical school students was administered using a tool that was adapted by Lee et al. [25]. This scale consists of a total of 2 items and is rated on a 4-point Likert scale. The total score range is 2–8 points, with 2–3, 4–5, and 6–8 points indicating low, normal, and high level of risk perception, respectively. The coefficient alpha in the study of Taghrir et al. [24] was 0.79, and was 0.67 in this study.

#### 2.3.3. Self-Efficacy

The Self-efficacy Scale used by Lee et al. [26] in a study on influenza preventive behavior was applied using a tool that was modified and supplemented by Zhang and Lee [27] in accordance with the COVID-19 situation. The scale comprised a total of 4 items on a 7-point Likert scale, and a higher score indicated a higher sense of self-efficacy. In the study of Zhang and Lee [27], the coefficient alpha of the scale was 0.80, and in this study, the coefficient alpha was 0.75.

#### 2.3.4. Prevention of COVID-19

Prevention of COVID-19 was measured using the COVID-19 prevention scale developed by Kim et al. [28] based on the Korea Centers for Disease guidelines for social distancing in daily life and the COVID-19 response guidelines. The scale consists of a total of 18 items, and the score is rated on a 4-point Likert scale; a higher core means a higher degree of prevention of COVID-19. In the study of Kim et al. [28], the coefficient alpha was 0.90, and in this study, it was 0.88.

### 2.4. Data Collection

The data collection for this study was undertaken from 1 December to 31 December 2021, and co-researcher posted an online notice inviting participant recruitment from those who wished to voluntarily participate in the research. The online questionnaire was provided through an URL to those who wanted to participate in the survey. The first display screen at the time of access was configured to include an explanation about the purpose, content, procedure, etc., of the research and documented consent to participate in the research. Only after agreeing to the research agreement could the respondent access the questionnaire, which took 10–15 min to complete.

### 2.5. Statistical Analyses

The data collected from the questionnaire were analyzed using the IBM SPSS/WIN 23.0 program (IBM Corporation, Armonk, NY, USA). The participant’s general characteristics, COVID-19 stress, risk perception, self-efficacy, and COVID-19 preventive behavior were analyzed using descriptive statistics. The Shapiro–Wilk normality test was performed, and then the difference in the prevention of COVID-19 according to general characteristics was analyzed using the *t*-test, analysis of variance, Mann–Whitney, and Kruskal–Wallis test. Correlations among COVID-19 stress, risk perception, self-efficacy, and COVID-19 preventive behavior were analyzed using Spearman correlation coefficient, and multiple linear regression analysis was performed for factors that affected the COVID-19 preventive behavior.

### 2.6. Ethical Considerations

This study was approved by the K University Research Ethics Review Committee (IRB no. 1040460-A-2021-045). The study was conducted through an online questionnaire survey, and the purpose, contents, and research procedure of the study were described and explained to all eligible participants. Only after consenting to participate via an online form could the participants access the survey on the next screen. The participants were assured that the related data would be processed anonymously, that any personal or identifiable information would not be exposed, and that the data would be used only for research purposes.

## 3. Results

### 3.1. Demographics

The general characteristics of participants are presented in Table 1. In this study cohort, there were 155 (86.6%) female and 24 (13.4%) male participants. With regard to the grade, respondents from the second year comprised the highest number of students (n = 59, 33.0%); 114 (63.7%) participants answered “yes” about the experience of education on respiratory infections, and 167 (93.3%) responded “yes” to the need for education on prevention of COVID-19; 134 (74.9%) participants had the experience of screening for COVID-19, and 45 (25.1%) did not have this experience. With regard to the subjective health status, “healthy” was the most frequent response (n = 94, 52.5%).

### 3.2. Degree of COVID-19 Stress, Risk Perception, Self-Efficacy, and COVID-19 Preventive Behavior

The nursing students’ COVID-19 stress score was 3.60 ± 0.57. Regarding sub-factors, anger towards others had the highest score of 3.97 ± 0.69, followed by fear of infection 3.70 ± 0.73 and difficulties due to social distancing 3.08 ± 0.80. Risk perception was 4.91 ± 1.26, self-efficacy was 5.92 ± 0.84, and the COVID-19 preventive behavior was 3.10 ± 0.44. Among sub-factors, prevention rules had the highest score of 3.59 ± 0.46, followed by personal hygiene rules 3.44 ± 0.46, high-risk facilities 3.06 ± 0.76, and distancing 2.58 ± 0.62 (Table 2).

### 3.3. Differences in the Preventive Behavior for COVID-19 According to Participant Characteristics

It was found that there were statistically significant differences in the prevention of COVID-19 according to sex (t = −2.40, *p* = 0.018) and grade (F = 3.52, *p* = 0.016) (Table 3).

### 3.4. Correlation among COVID-19 Stress, Risk Perception, Self-Efficacy, and COVID-19 Preventive Behavior

The nursing students’ COVID-19 preventive behavior showed a significantly positive correlation with COVID-19 stress (r = 0.51, *p* < 0.001), risk perception (r = 0.21, *p* = 0.006), and self-efficacy (r = 0.62, *p* < 0.001) (Table 4).

### 3.5. Influencing Factors on COVID-19 Preventive Behavior

Multiple linear regression analysis was performed to identify the factors that affected the nursing students’ COVID-19 prevention behavior. The Durbin–Watson statistic was 1.81; as this was close to 2, there was no autocorrelation problem. The tolerance was 0.542–0.852, which was larger than 0.1, and the variance inflation factor (VIF) was 1.331–1.846, and as it did not exceed 10, there was no problem of multicollinearity. Therefore, the assumption for multiple linear regression analysis was satisfied. Among the general characteristics, sex and grade, which showed a significant difference in the COVID-19 preventive behaviors, were treated as dummy variables and input, and COVID-19 stress, risk perception, and self-efficacy variables were input. The results of the analysis showed that the factors that significantly affected the COVID-19 preventive behaviors were self-efficacy (β = 0.53, *p* < 0.001) and COVID-19 stress (β = 0.25, *p* = 0.001), while sex, grade and risk perception had no significant effect. The explanatory power of the variable was 45% (F = 22.03, *p* < 0.001) (Table 5).

## 4. Discussion

This study was performed to confirm the association of COVID-19 stress, risk perception, and self-efficacy with COVID-19 preventive behavior of nursing students, and to identify the factors that affect COVID-19 preventive behavior.

The degree of COVID-19 preventive behavior of the nursing students was 3.10 ± 0.44 (range: 1–4); in the sub factors, prevention rules showed the highest score. The survey was conducted in December 2021, when the prevention rules had been strengthened, such as in the reduction in the number of people at private gatherings and the expansion of facilities where the quarantine pass was applicable. Among the questionnaire items, it was confirmed that the items related to prevention rules, such as “cover your nose and mouth completely when wearing a mask” and “do not go out and avoid contact with others if you have a fever or respiratory symptoms” were high, which was confirmed by the government’s high score. This high score is a reflection of the strengthening of prevention rules. Additionally, the items related to distance showed low scores. Actions such as keep a distance of 2 m in line or sit in a zigzag while eating were not observed well; thus, it is necessary to increase interest in distancing to improve the performance of COVID-19 preventive behavior.

Although the survey period and tools are different, in the study of Kwak and Kim [29], which investigated the infection preventive behavior of nursing students, the score was 4.58 ± 0.43 (range: 1–5), and in the study of Park et al. [15], the score was 4.44 ± 0.47 (range: 1–5). Therefore, we confirmed that nursing students had a high performance of preventive measures against COVID-19.

The COVID-19 preventive behavior showed significant differences according to gender and grade. In the analysis stratified by sex, female students had high scores in infection preventive behavior, and in a study with nursing students [15] and another with medical students [24], female students showed a high degree of infection prevention, although there was no significant difference. In general, the ratio of female nursing students was high, and the COVID-19 preventive behavior of nursing students was higher than those of other majors [30]. Since this study examined the degree of COVID-19 preventive behavior of nursing student within one university, the relationship between sex and COVID-19 preventive behavior of nursing students should be examined in future studies, in consideration of various regions and characteristics. In this study, the COVID-19 preventive behavior was highest among the fourth graders, similar to the finding by Park et al. [31]. The high level of infection prevention behavior by the fourth graders could be a result of acquiring direct and indirect experiences about the risk of infection, as well as actual practice of various infection prevention behaviors learned in the lower grades. In addition, the effectiveness of infection prevention-related education acquired in nursing shows a gradual increase, which implies that the degree of performing the COVID-19 preventive behavior is highest in the fourth grade.

The results of this study showed that the factors affecting the COVID-19 preventive behaviors were self-efficacy and COVID-19 stress. In a study [19] among university students, self-efficacy was found to be a significant influencing factor for preventive health behaviors against emerging infectious diseases, which the results of this study support. Self-efficacy is a strong factor that causes behavioral change, and when self-efficacy is high, efforts to solve problems increase [17]. The results of this study showed that self-efficacy was positively correlated with COVID-19 stress, and this was also supported by a study [20] among older individuals living at home. A high self-efficacy can be interpreted as a high determination to solve problems more actively to overcome stressful situations [20]. Therefore, it is expected that the implementation of COVID-19 preventive behavior can be increased by enhancing self-efficacy, and it is necessary to develop various programs to improve self-efficacy for nursing students. Self-efficacy can be enhanced through achievement experience, vicarious experience, verbal persuasion, and emotional awakening [17]. Thus, a self-efficacy enhancement program that can improve the prevention of infectious diseases should be implemented.

Furthermore, this study confirmed that COVID-19 stress was a significant influencing factor in infection preventive behavior. In a study [9] of some college students, excluding health departments, COVID-19 stress was found to be a significant influencing factor on COVID-19-related preventive health behavior, which is supported by the results of this study. However, COVID-19 preventive behaviors can actually increase stress. Due to the COVID-19 pandemic, social distancing, self-isolation, and restrictions on outside activities have been experienced, and the prolonged COVID-19 pandemic has had a serious impact on the mental health of young people in various aspects, such as job instability and disruption of education and training [32]. In this study, COVID-19 stress was 3.60 ± 0.57 (range: 1–5), and in the study [33] with nursing students, the stress score was 3.28 ± 0.89 (range: 1–5); moreover, in the study of older individuals living at home [20], the score was 1.63 ± 0.69 (range: 0–4), which confirms that young people have a high level of COVID-19 stress. A previous study [9] found that COVID-19-related stress is high, but infection prevention is performed to comply with the government’s quarantine rules. The results of this study can be interpreted as expecting a quick recovery to daily life through compliance with quarantine regulations despite the increase in stress caused by the prolonged COVID-19 pandemic.

The results of this study showed that the score for “Anger toward others” was the highest among the sub-factors of COVID-19 stress. This consists of questions related to anger at other people not following the quarantine rules properly, and it can be understood in a similar context to how young people express their anger at the spread of infection through the social activities of their peers [34]. Additionally, among the sub-factors of COVID-19 stress, the question about fear was found to be at a high level. It consists of questions about anxiety about COVID-19 infection, fear of after-effects of COVID-19, and pain when infected with COVID-19. When you feel the severity or risk of COIVD-19 infection, it can be seen that you perform more preventive behaviors.

Since COVID-19 stress affects depression, social isolation, and college life adjustment [33,35], various efforts are needed to alleviate the negative situations caused by COVID-19 stress. In particular, nursing students had negative experiences, such as suspension of clinical practice, during the spread of COVID-19. Therefore, in addition to COVID-19 stress, it is necessary to think about various factors that can affect nursing students’ COVID-19 prevention behavior.

The results of this study have shown that risk perception is not a significant influencing factor on the COVID-19 preventive behavior. In a study [25] of nursing students that used the same research tool, risk perception was found to be a significant influencing factor on the prevention of COVID-19, showing contradictory results with the results of this study, while another study [16], risk perception was not found to be a significant influencing factor, and is supported by the results of this study. In this study, the risk recognition tool had low reliability, and it is thought that the score for risk perception may be affected by the degree of COVID-19 epidemic. Therefore, in future studies, it is necessary to check the degree of risk perception by considering other risk recognition tools and timing of measurement. In this study, risk perception was average, which is consistent with a previous study [25] using the same tool. In the early stages of the infectious disease, awareness of the risk of COVID-19 and prevention of infection were low. However, as the disease spread, awareness of the risk of COVID-19 gradually increased [36]. As providing information on infectious diseases can instill an accurate awareness of the risk of infection [37], increased risk awareness through effective information delivery on COVID-19 is necessary to promote infection preventive practices [14]. This study showed that risk perception has a significant correlation with COVID-19 stress and COVID-19 preventive behavior; therefore, it is necessary to ascertain the relationship between risk perception and COVID-19 preventive behavior through future research.

This study examined the COVID-19-related stress, risk perception, self-efficacy, and the degree of COVID-19 preventive behavior of nursing students during the prolonged COVID-19 epidemic and provides basic data for developing an interventional program to promote COVID-19 preventive behavior, which makes this study meaningful. Through this intervention, nursing students will contribute to improving the practice rate of infection prevention in the event of a new infectious disease in the future in their role as preliminary medical personnel. As the results of this study were obtained by convenience sampling of nursing students at one regional university, the results may not be generalizable. Thus, follow-up studies considering various regional and departmental characteristics are needed in the future. In addition, since research findings may vary depending on the epidemic period, and it is impossible to determine the impact of COVID-19 prevention behavior on the spread of COVID-19, a longitudinal study may be necessary. Further, future studies should evaluate additional variables that may influence COVID-19 prevention behavior. Lastly, based on the results of this study, to promote the COVID-19 preventive behaviors, it is necessary to improve the self-efficacy of nursing students and manage COVID-19 stress according to preventive behaviors.

## 5. Conclusions

This study was performed to identify factors that affect the COVID-19 preventive behavior of nursing students. Self-efficacy and COVID-19 stress were identified as significant influencing factors in nursing students’ COVID-19 preventive behavior, and the total explanatory power of these variables was 45%. Based on the results of this study, it is necessary to develop an interventional program for promoting self-efficacy related to COVID-19 and appropriate stress management to improve the COVID-19 preventive behavior of nursing students.

## Figures and Tables

**Table 1 ijerph-19-12094-t001:** General characteristics of participants.

Characteristics	Categories	N (%)
Sex	Male	24 (13.4)
	Female	155 (86.6)
Grade	1st	47 (26.3)
	2nd	59 (33.0)
	3rd	47 (26.3)
	4th	26 (14.5)
Education experience	Yes	114 (63.7)
about Respiratory Infection	No	65 (36.3)
Necessity of COVID-19	Yes	167 (93.3)
prevention education	No	12 (6.7)
COVID-19 testing experience	Yes	134 (74.9)
	No	34 (25.1)
Subjective health status	Very healthy	43 (24.0)
	Healthy	94 (52.5)
	Moderate	39 (21.8)
	Unhealthy	3 (1.7)

Note: COVID-19: coronavirus disease.

**Table 2 ijerph-19-12094-t002:** Degrees of COVID-19 stress, risk perception, self- efficacy and COVID-19 preventive behavior.

Variables	Item M ± SD or M ± SD	Range
COVID-19 stress	3.60 ± 0.57	1–5
Fear of infection	3.70 ± 0.73	
Difficulties due to social distancing	3.08 ± 0.80	
Anger towards others	3.97 ± 0.69	
Risk perception	4.91 ± 1.26	2–8
Self-efficacy	5.92 ± 0.84	1–7
COVID-19 preventive behavior	3.10 ± 0.44	1–4
Distancing	2.58 ± 0.62	
Prevention rules	3.59 ± 0.46	
Personal hygiene rules	3.44 ± 0.46	
High-risk facilities	3.06 ± 0.76	

Note: M ± SD: mean ± standard deviation; COVID-19: coronavirus disease.

**Table 3 ijerph-19-12094-t003:** Difference in COVID-19 preventive behavior by general characteristics.

Characteristics	Categories	COVID-19 Preventive Behavior	
		M ± SD	t /F/z (*p*)
Sex	Male	2.90 ± 0.57	−2.40 (0.018)
	Female	3.13 ± 0.41	
Grade	1st	2.94 ± 0.52	3.52 (0.016)
	2nd	3.18 ± 0.41	
	3rd	3.08 ± 0.36	
	4th	3.22 ± 0.42	
Education experience about	Yes	3.14 ± 0.39	1.68 (0.097)
Respiratory Infection	No	3.03 ± 0.51	
Necessity of COVID-19	Yes	3.11 ± 0.43	−1.69 (0.092) *
prevention education	No	2.93 ± 0.52	
COVID-19 testing experience	Yes	3.10 ± 0.45	0.14 (0.886)
	No	3.10 ± 0.41	
Subjective health status	Very healthy	3.06 ± 0.51	2.28 (0.517) ^†^
	Healthy	3.14 ± 0.40	
	Moderate	3.02 ± 0.45	
	Unhealthy	3.15 ± 0.33	

Note: M ± SD: mean ± standard deviation, COVID-19: corona virus disease, * Mann–Whitney test, ^†^ Kruskal–Wallis test.

**Table 4 ijerph-19-12094-t004:** Correlation among COVID-19 stress, risk perception, self-efficacy, and COVID-19 preventive behavior.

Variables	COVID-19 Stress	Risk Perception	Self-Efficacy
	r (*p*)	r (*p*)	r (*p*)
Risk perception	0.41 (<0.001)	1	
Self-efficacy	0.42 (<0.001)	0.05 (0.505)	1
COVID-19 preventive behavior	0.51 (<0.001)	0.21 (0.006)	0.62 (<0.001)

Note: COVID-19: coronavirus disease.

**Table 5 ijerph-19-12094-t005:** Influencing factors on COVID-19 preventive behavior.

Variables		B	S.E	β	t	*p*
(Constant)		0.17	0.24	-	0.69	0.488
Sex * (ref = male)	female	−0.14	0.09	−0.09	−1.47	0.143
Grade * (ref = 1st)	2nd	0.06	0.08	0.05	0.71	0.481
	3rd	0.03	0.09	0.03	0.36	0.717
	4nd	0.07	0.10	0.05	0.70	0.486
Self-efficacy		0.34	0.04	0.53	7.76	<0.001
Risk perception		0.02	0.03	0.05	0.71	0.478
COVID-19 stress		0.23	0.07	0.25	3.25	0.001
R^2^ = 0.47 Adjusted R^2^ = 0.45 F = 22.03 *p* < 0.001						

Note: COVID-19: coronavirus disease, S.E: standard errors, * dummy variables.

## Data Availability

The data sets used and/or analyzed during the present study are available from the corresponding author on reasonable request.

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
