# Peer review of "Factors Influencing Preventive Behavior of COVID-19 among Nursing Students in South Korea"

_ijerph, 2022, doi:10.3390/ijerph191912094_

Round 1
Reviewer 1 Report
Thank you for allowing me to read your manuscript. I wonder why you have not included age in the demographic data, as age has been shown to be a significant indicator of stress and anxiety in relation to COVID-19. Also I wonder whether you thought about balancing the gender split, or even removing the males from the study, as these are a significant minority in your participation pool. I also wonder whether you could concentrate on the regression analysis rather than doing a scattergun approach to the statistical analysis and having your focus on one particular analysis. At the moment the paper reads a little disjointed in regards to the amount of analysis that you have done that aren't necessarily justified. I think your findings are interesting and are in parallel with other studies of a similar nature but I'm not sure that your study is particularly unique. If the authors could identify a unique aspect this would enhance the current manuscript. I also think that you need to justify why you used nurses as opposed to any other university students in much more detail you have given some detail but the justification is a little bit weak at the moment.
Reviewer 2 Report
Please see attached

Round 2
Reviewer 1 Report
Even though I still have concerns regarding the scattergun approach to the statistics, you have addressed the vast majority of my concerns.
Author Response
Next time, we will try to become a more complete journal in consideration of the comments made by the judges. Additionally, the manuscript was reviewed in consideration of the reviewers' comments. thank you.
Reviewer 2 Report
Suggested changes have been made. I'm still skeptical that increasing self efficacy and reducing stress will improve COVID-19 protective behaviors (association does not imply causation). I also wonder if we can really improve self efficacy and reduce stress during a really stressful time.
Author Response
Additionally, the manuscript was reviewed in consideration of the reviewers' comments.
We thank the reviewer for the comment. We have modified the text as follows:
Also, among the sub-factors of COVID-19 stress, the question about fear was found to be at a high level. It consists of questions about anxiety about COVID-19 infection, fear of after-effects of COVID-19, and pain when infected with COVID-19. When you feel the severity or risk of COIVD-19 infection, it can be seen that you perform more preventive behaviors.